# Host Response of Syrian Hamster to SARS-CoV-2 Infection including Differences with Humans and between Sexes

**DOI:** 10.3390/v15020428

**Published:** 2023-02-03

**Authors:** Martina Castellan, Gianpiero Zamperin, Giulia Franzoni, Greta Foiani, Maira Zorzan, Petra Drzewnioková, Marzia Mancin, Irene Brian, Alessio Bortolami, Matteo Pagliari, Annalisa Oggiano, Marta Vascellari, Valentina Panzarin, Sergio Crovella, Isabella Monne, Calogero Terregino, Paola De Benedictis, Stefania Leopardi

**Affiliations:** 1Division of Comparative Biomedical Sciences, Istituto Zooprofilattico Sperimentale delle Venezie, 35020 Legnaro, Italy; 2Laboratory for Emerging Viral Zoonoses, Istituto Zooprofilattico Sperimentale delle Venezie, 35020 Legnaro, Italy; 3Viral Genomics and Transcriptomics Laboratory, Istituto Zooprofilattico Sperimentale delle Venezie, 35020 Legnaro, Italy; 4Laboratory of Diagnostic Virology, Istituto Zooprofilattico Sperimentale della Sardegna, 07100 Sassari, Italy; 5Laboratory of Histopathology, Istituto Zooprofilattico Sperimentale delle Venezie, 35020 Legnaro, Italy; 6Risk Analysis and Public Health Department, Istituto Zooprofilattico Sperimentale delle Venezie, 35020 Legnaro, Italy; 7Innovative Virology Laboratory, Istituto Zooprofilattico Sperimentale delle Venezie, 35020 Legnaro, Italy; 8Laboratory of Experimental Animal Models, Istituto Zooprofilattico Sperimentale delle Venezie, 35020 Legnaro, Italy; 9Biological Science Program, Department of Biological and Environmental Sciences, College of Arts and Sciences, Qatar University, Doha P.O. Box 2713, Qatar

**Keywords:** SARS-CoV-2, animal model, sex, histopathology, host response, immune response, transcriptomic profile

## Abstract

The emergence of severe acute respiratory syndrome coronavirus 2 (SARS-CoV-2) has highlighted the importance of having proper tools and models to study the pathophysiology of emerging infectious diseases to test therapeutic protocols, assess changes in viral phenotypes, and evaluate the effects of viral evolution. This study provided a comprehensive characterization of the Syrian hamster (*Mesocricetus auratus*) as an animal model for SARS-CoV-2 infection using different approaches (description of clinical signs, viral load, receptor profiling, and host immune response) and targeting four different organs (lungs, intestine, brain, and PBMCs). Our data showed that both male and female hamsters were susceptible to the infection and developed a disease similar to the one observed in patients with COVID-19 that included moderate to severe pulmonary lesions, inflammation, and recruitment of the immune system in the lungs and at the systemic level. However, all animals recovered within 14 days without developing the severe pathology seen in humans, and none of them died. We found faint evidence for intestinal and neurological tropism associated with the absence of lesions and a minimal host response in intestines and brains, which highlighted another crucial difference with the multiorgan impairment of severe COVID-19. When comparing male and female hamsters, we observed that males sustained higher viral RNA shedding and replication in the lungs, suffered from more severe symptoms and histopathological lesions, and triggered higher pulmonary inflammation. Overall, these data confirmed the Syrian hamster as a suitable model for mild to moderate COVID-19 and reflected sex-related differences in the response against the virus observed in humans.

## 1. Introduction

The pandemic of coronavirus disease 2019 (COVID-19) has resulted in a devastating global threat to human society, the economy, and the healthcare system [1,2,3]. The disease is caused by the severe acute respiratory syndrome coronavirus 2 (SARS-CoV-2), a positive-sense single-stranded RNA virus that belongs to the subgenus *Sarbecovirus*, genus *Betacoronavirus*, and species *SARS-related coronavirus* that emerged from an animal source after zoonotic cross-species transmission [4,5,6]. The virus mostly replicates in the respiratory tract, although patients may also experience disorders associated with multiorgan engagement that include neurological and gastroenteric symptoms, the incidence, mechanism, and significance of which are still a matter of discussion [7,8,9,10,11]. According to the age of the patient and the presence of predisposing factors, COVID-19 varies widely in the severity of its clinical manifestations and spans from asymptomatic infections to an acute respiratory distress syndrome (ARDS) that requires mechanical ventilation and, in the worst-case scenario, leading to death [12,13,14]. Epidemiological data indicate that males are more prone to develop a severe COVID-19 symptomatology, which suggests that sex may also influence SARS-CoV-2 pathogenesis due to genetic and hormonal factors, although social–behavioral differences between genders may also play a role [15,16,17,18]. Regardless of the cause, the severe form of the disease follows a common mechanism in the dysregulation of the inflammatory response, which is similar to what has been observed in other coronavirus infections such as severe acute respiratory syndrome (SARS) and Middle East respiratory syndrome (MERS) [12]. Briefly, in the attempt to clear the infection, the immune system of certain individuals releases an excessive amount of pro-inflammatory cytokines known as a “cytokine storm” that promotes an uncontrolled inflammation that damages lungs and other organs such as the brain, gut, and heart [19,20]. This important evidence has paved the way toward a diagnostic, prognostic, and therapeutic approach that is focused on controlling patients’ immune responses with particular attention paid to the innate immunity [20,21]. The overarching goal is to control the pandemic by reducing the incidence of severe manifestations through vaccination campaigns and to develop and assess the efficacy of therapeutic agents against the new variants derived from the evolution of SARS-CoV-2. Among these, the WHO classifies as “variants of concern (VOCs)” viruses [22] that show mutations on the spike protein that might influence transmissibility, symptomatology, immune evasion, efficacy of therapeutic monoclonal antibodies, and sensitivity of diagnostic methods [23,24,25,26,27,28,29,30]. The first variant defined as VOC was the Alpha (also referred to as B.1.1.7), which, compared to older strains, was associated with higher transmissibility and, according to some studies, increased mortality rates [31]. The Alpha VOC, which was first described in the United Kingdom in November 2020, rapidly spread across Europe and was responsible for an increased number of infections during the second epidemic wave. In Italy, it was the most prevalent variant between February and March 2021 [32,33].

In the race against a pandemic event, researchers need reliable animal models that (i) are susceptible to the infection, (ii) are able to eliminate the virus, (iii) display clinical and pathological manifestations typical of the human disease, and (iv) mimic the same immune impairments found in patients. Non-human primates, ferrets, hamsters, and transgenic mice (i.e., K18-hACE2 mice) are susceptible to SARS-CoV-2 infection and develop lung lesions that resemble the pathological patterns found in humans [31,34,35,36,37,38,39]. Among these, the Syrian hamster (*Mesocricetus auratus*) exhibits the best balance between costs, neurological development, availability, easy handling, and maintenance in captivity, so it is extensively used for translational medicine [31,36,40,41]. Previous studies showed that SARS-CoV-2 replicated efficiently in the respiratory tract of hamsters and was able to invade the central nervous system with no differences observed between animals of different ages [40,42]. Histopathological and radiographic evaluations confirmed that these animals developed pneumonia without showing severe clinical manifestations and fully recovered in 2–3 weeks, although chronic sequelae could be seen in the lungs, kidney, and the olfactory bulb up to 31 days after infection [40,42,43]. In addition, preliminary studies showed that hamsters increase the gene expression of some cytokines/chemokines in the lungs, which may be compatible with the cytokine storm described in humans [42,44]. The aim of this study was to provide an in-depth evaluation of the Syrian hamster as animal model for human COVID-19 and to identify the advantages and disadvantages of using this species for translational medicine. Our work provided new outcomes that need to be taken into account while designing an infection study that uses this animal model, including evidence for sex-related peculiarities and critical differences with the human host response.

## 2. Materials and Methods

### 2.1. Animal Experiment

The study involved 60 8-week-old Syrian hamsters that were divided into 4 experimental groups of 15 individuals, each of which was composed of only females or only males that were either infected or used as negative controls. Animals were acclimatized 7 days prior to infection in individual cages (BCU-2 Rat Sealed Negative Pressure IVC, Allentown Inc, Allentown, NJ, USA) in the biosafety level 3 (BSL3) facility following national and international regulations on the welfare of laboratory animals.

Animals in the infected groups were inoculated intranasally under general anesthesia with isoflurane using the SARS-CoV-2 B.1.1.7 (Alpha) variant (accession no.: EPI_ISL_766579) isolated in 2020 from a mild/severe COVID case as previously described [45]. Briefly, the virus was isolated in Vero E6 cells (ATCC^®^ CRL 1586 ™) and maintained in Dulbecco’s modified Eagle’s medium (DMEM, Thermo Fisher, Waltham, MA, USA) supplemented with 10% fetal calf serum (FCS), penicillin (100 U/mL), and streptomycin (100 U/mL) (all from Thermo Fisher). Viral inoculates used for the animal experiment were produced by infecting Vero E6 cells (DMEM supplemented with 2% FCS, penicillin (100 U/mL), and streptomycin (100 U/mL)) at a multiplicity of infection (MOI) of 1 with the second culture passage of the original isolate. We inoculated 100 µL of the purified cell suspension with a viral dose of 8 × 10^4^ PFU, which was within the range used in the literature and is known to induce productive infection in this model [40,43,46]. Hamsters included in the two groups of negative controls were inoculated using 100 μL of sterile PBS solution [40,42,43,47].

Animals were monitored daily for 14 days to record clinical signs and general signs of distress, thus securing animal welfare in addition to collecting research data. At 2, 4, 6, 9, and 14 dpi, we registered weights and collected oropharyngeal swabs and blood samples from the gingival vein under general anesthesia [48]. On days 2, 6, and 14, we euthanized 5 individuals per group and performed an intra-cardiac terminal blood collection for PBMCs isolation with Ficoll-Paque Plus (GE healthcare). We performed a complete necropsy of all animals and collected samples of the lungs, brains, and intestines using the best practices to avoid cross-contamination between different organs and animals. Specimens were fixed in 10% neutral-buffered formalin and in RNA later (Thermo Fisher^®^, Waltham, MA, USA) for histological examination and molecular analyses, respectively (Appendix A). Further details on tissue collection can be found in the Appendix A.

### 2.2. Histology and Immunofluorescence

Formalin-fixed samples were embedded in paraffin, cut in 4 µm-thick sections, and stained with hematoxylin and eosin (H&E) to evaluate the presence and severity of lesions in different organs. Histopathological lesions were scored as described elsewhere [44] (Appendix A); the total histopathological score was defined as the sum of the scores attributed to each specific lesion. Slides were analyzed and images were taken using a Leica DM4 B light microscope with a DFC450 C Microscope Digital Camera at 20X and the software Leica Application Suite V4.13 (Leica Microsystems, Wetzlar, Germany).

We investigated the presence of the virus within tissues by immunofluorescence by using anti-SARS-CoV-2 spike glycoprotein and anti-dsRNA [49,50] as primary antibodies in order to discriminate the presence of the antigen by the active replication of the virus within tissues; this was based on the fact that dsRNA is widely known as a replicative intermediate for coronaviruses [51]. Immunofluorescence was also applied to investigate the expression of the ACE2 receptors within the tissues. Further details can be found in the Appendix A.

### 2.3. Molecular Analyses for SARS-CoV-2 Detection and Quantification

The presence of viral RNA in oropharyngeal swabs was detected in all of the control and infected individuals via qualitative rRT-PCR using the AgPath-ID ™ One-Step RT-PCR Reagents (Life Technologies) on a CFX96 Touch Deep Well Real-time PCR Detection System (Biorad). To quantify SARS-CoV-2 in the target organs, we developed a digital droplet RT-PCR (RT-ddPCR) that employed the One-Step RT-ddPCR Advanced Kit for Probes (Bio-Rad) and the QX200 Droplet Digital PCR System (Biorad). This approach was implemented only for the three individuals per experimental group that were randomly selected for transcriptomic analyses. Quantitative data were expressed as Log2 copies/mL of genomic RNA. Both tests targeted the SARS-CoV-2 envelope protein (E) gene [52]. The quality of the samples was verified by amplification of the β-actin mRNA [53]. Further details can be found in the Appendix A.

### 2.4. Gene Expression Analyses by RNA-Seq

We randomly selected three individuals among five of each experimental group to investigate the virus–host response by performing the transcriptomic profile of the lungs, brains, intestines, and PBMCs of infected versus mock animals at three different time points along the infection: early infection, the infection apex, and recovery.

Libraries were prepared using the Truseq Stranded mRNA library preparation kit (Illumina) following the manufacturer’s instructions and were run on an Agilent 2100 Bioanalyzer using an Agilent High Sensitivity DNA kit (Agilent Technologies) to ensure the proper range of cDNA length distribution. Sequencing was performed on an Illumina NextSeq with a NextSeq^®^ 500/550 High Output Kit v2.5 (300 cycles; Illumina, San Diego, CA, USA) in pair-end [PE] read mode, which produced about 33 million reads per sample. After filtering the raw data, we aligned the high-quality reads against the reference genome of the *Mesocricetus auratus* (BCM Maur 2.0, NCBI) [54] using STAR v2.7.9a [55] and generated the gene count using htseq-count v0.11.0 [56]. We then investigated the differential expression of genes between infected and mock males and females at each time point using the Deseq2 package [57] and assigned Gene Ontology (GO) terms to each gene using Blast2GO v5.2.5 [58]. Child–father relationships that belonged to the GO graph were reconstructed using the OBO file downloaded from http://geneontology.org (accessed on 19 October 2021). Orthologs with *Homo sapiens*, *Mus musculus*, and *Rattus norvegicus* were computed using Orthofinder v2.5.4 [59], and their proteomes were downloaded from Ensembl. Further details can be found in the Appendix A.

### 2.5. Serological Analyses

In order to evaluate the sero-conversion dynamics, we performed the focus reduction neutralization test (FRNT) as previously described using the same viral strain used for the infection for the detection of neutralizing antibodies [60,61]. We defined the serum neutralization titer as the reciprocal of the highest dilution that resulted in a reduction in the control focus count higher than 90% (FRNT90). Sera of all animals were collected at 2, 4, 6, 9, and 14 dpi; only 4 out of 5 sera were collected for both males and females at 9 dpi.

We further analyzed the serum samples of all the controls and infected animals at 2, 4, 6, 9, and 14 dpi for the presence of pro-inflammatory cytokines. Levels of IL-1β and IL-6 were assessed at the Istituto Zooprofilattico Sperimentale della Sardegna via singleplex ELISA using target-specific ELISA kits (MyBiosource), according to the manufacturers’ instructions and using an Epoch microplate reader (BioTek) to read the absorbance.

### 2.6. Statistical Analyses

We adopted the minimum sample size that guaranteed an effective comparison between groups while minimizing the use of experimental animals. Infection of 13 out of 15 individuals per group indicated successful infection with a first type error α = 0.01 (one tail) and a power 1-β = 0.85.

To describe the shedding of viral RNA in males and females and to monitor the hamsters’ weights during infection of males and females versus the corresponding group of mock animals, we performed a descriptive statistical analysis using SAS 9.4 software [62,63,64]. We applied a spline mixed model according to sex while taking into account the correlation between different observations of the same hamster over time using a first-order autoregressive (AR(1)) structure for the covariance matrix. We applied a type III F-test to evaluate the significance of the overall effect of the fixed factors specified in the model and performed post hoc pairwise comparisons for each fixed factor of the mixed models to clarify differences. We performed a studentized residuals analysis to assess the quality of the model and plotted a Q–Q plot, the residuals’ distribution, and a scatterplot of the residuals versus the fitted to verify the assumption of normality and homoscedasticity; *p*-values of <0.05 were considered significant (including the Sidak-adjusted *p*-values provided for multiple tests). For all the remaining statistical analyses, we used the Wilcoxon–Mann–Whitney test for independent groups implemented in GraphPad Prism 9. These included comparisons of: (i) the median antibody titers and serum levels of IL-1β and IL-6 of females versus males at each time point; (ii) histopathological total scores between infected versus mock animals of each sex and between infected females vs. infected males at 2, 6, and 14 dpi; and (iii) intra-alveolar inflammatory cell infiltration and perivascular/alveolar edema histopathological scores (the lesions were more abundant in males; see the Results section) between infected versus mock animals of each sex and between infected females vs. infected males at 2, 6, and 14 dpi. For all statistics, we considered *p*-values < 0.05 as significant.

### 2.7. Data Availability

The RNA-Seq raw data generated in the present study were deposited in SRA under accession number PRJNA839918. The raw data for Figure 1a,b, Figure 2d–f, Figure 3a–c, Figure 4a–c, Figure 5a,b and Figure 6a,c are provided as Appendix A.

## 3. Results

### 3.1. Infection and Seroconversion

Syrian hamsters that were intranasally infected with the B.1.1.7 SARS-CoV-2 VOC developed no clinical signs except for a 5% drop in body weight between 2 and 6 dpi with subsequent recovery (Figure 1a; Appendix A for weight values per sex, status, and days after infection). The shedding of viral RNA began at 2 dpi, peaked between 4 and 6 dpi depending on the sex, and dropped shortly after. The virus genome was detectable until 14 dpi with high CT values; males showed higher shedding across the entire study period (*p* < 0.0001; Figure 1b) with a mean delta of 3.6 CT (Figure 1b; Appendix A).

All infected individuals produced detectable neutralizing antibodies against SARS-CoV-2 starting at 6 dpi and reached the highest titers at 14 dpi (Figure 1c). Geometric mean titers (GMTs) were higher in males rather than females, but the difference was not statistically significant.

SARS-CoV-2 established a productive infection in the lungs, and viral RNA was detected in all individuals with a decreasing viral load over time (Figure 1d). We confirmed these results by showing the presence of the spike protein in the pulmonary parenchyma of all the individual by using immunofluorescence. Interestingly, the lungs of male hamsters collected at 6 dpi also showed a marked expression of double-strand RNA (dsRNA), which is an indicator of active viral replication, whereas no signal could be observed in the lungs of the females at the same time point (Figure 1e). The mock animals did not stain for any of the tested antibodies, which confirmed the specificity of the reactions. Evidence of SARS-CoV-2 infection in the intestines and brains was far less marked: there were low viral loads and inconsistent results within the infected groups. On day 14, only one individual was positive in each group in both organs (Figure 1d). Coherently with the molecular results, immunofluorescence staining for the viral spike glycoprotein and dsRNA was evident only in the intestinal sections of two males at 2 dpi (Appendix A), and no signal was detected in infected brains at any of the time points analyzed (Appendix A).
Figure 1SARS-CoV-2 infection in oropharyngeal swabs, lungs, and distal organs. (**a**) Statistical model describing body weight over time in mock and infected male (Mock M, M) and female (Mock F, F) Syrian hamsters. (**b**) Statistical model describing rRT-PCR results trend of RNA extracted from oropharyngeal swabs at 2, 4, 6, 9, and 14 dpi. (**a**,**b**): infected and control females (2 dpi n = 30; 4 dpi n = 20; 6 dpi n = 20; 9 dpi n = 10; 14 dpi n = 10) and infected and control males (2 dpi n = 30; 4 dpi n = 20; 6 dpi n = 20; 9 dpi n = 10; 14 dpi n = 10). See Appendix A for further details. (**c**) Focus reduction neutralization test (FRNT) results expressed as the reciprocal of the highest dilution that resulted in a reduction in the control focus count >90% (FRNT90). Geometric mean titers (GMTs) with 95% confidence intervals (CI) are represented. Dotted lines indicate the limit of detection (LOD). Wilcoxon–Mann–Whitney test of males vs. females: 2 dpi *p* > 0.99; 4 dpi *p* > 0.99; 6 dpi *p* = 0.16; 9 dpi *p* = 0.63; 14 dpi *p* = 0.59. Infected females: 2 dpi n = 5; 4 dpi n = 5; 6 dpi n = 5; 9 dpi n = 4; 14 dpi n = 5. Infected males: 2 dpi n = 5; 4 dpi n = 5; 6 dpi n = 5; 9 dpi n = 4; 14 dpi n = 5. (**d**) SARS-CoV-2 viral load as determined via RT-ddPCR in the lungs, intestines, and brains at 2, 6, and 14 dpi; results are expressed as Log2 copies/mL of genomic RNA for graphical comparison among organs. Infected female and male lungs and intestines: 2, 6, and 14 dpi n = 3 each; infected female and male brains: 2 dpi n = 3; 6 dpi n = 2; 14 dpi n = 3 each. No statistical analyses were performed on this dataset due to the low sample size. (**e**) Representative immunofluorescence staining for SARS-CoV-2 spike glycoprotein (green) and dsRNA (red) in infected male and female lungs. Scale bar = 25 μm. All animals were analyzed; representative images are shown.
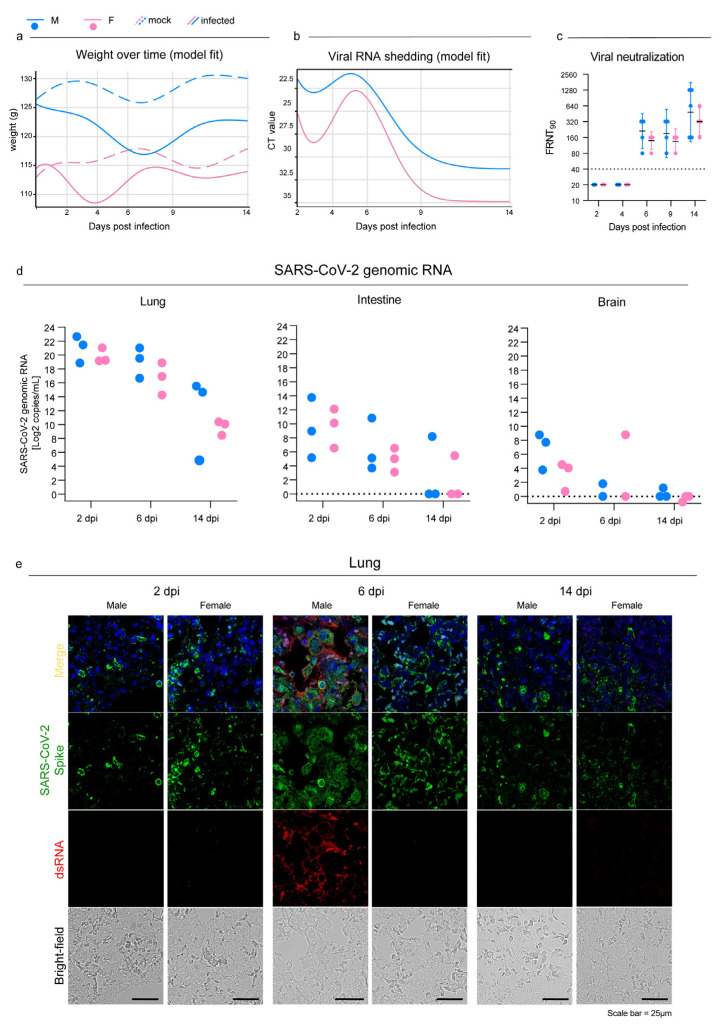



### 3.2. Pathology

Macroscopically, the lungs of the male hamsters were diffusely consolidated with a dark-red coloration on day 6, while multiple dark-red consolidated areas were scattered throughout all the lobes of the females. At 14 dpi, we observed few small reddish foci independently of the sex.

Histopathogical changes in the lungs were consistent with a bronchointerstitial pneumonia (Figure 2a–c); the total histopathological score peaked on day 6 in both sexes (Figure 2d; Appendix A). At 2 dpi, the main histopathological changes consisted of mild to moderate alveolar damage (with infiltrates of macrophages and few neutrophils) and vascular hyperemia (Figure 2a). At 6 dpi, extensive and coalescing inflammatory foci with parenchymal consolidation affected more than 75% of the surface in three individuals, 50–75% in five, and 25–50% in two females. In all animals, alveolar damage was associated with intense pneumocyte type II and bronchiolar epithelium hyperplasia (Figure 2b and Figure 2g.1). We detected scattered syncytial multinucleated cells in bronchioles and alveolar surfaces that in one case contained 2–4 µm amphophilic round cytoplasmic inclusions that were consistent with viral inclusion bodies (Figure 2g.2). At this stage, edema and infiltration of inflammatory cells (perivascular lymphomonocytic cuffs, alveolar macrophages, and neutrophils) were moderate to severe and slightly more abundant in males (Figure 2e,f and Appendix A). Scattered fibrin exudation in alveolar lumina was detected, and there was no hyaline membrane formation. Pre- and post-capillary vasculature exhibited plumped reactive endothelium with a sub-endothelial infiltration of lymphocytes, monocytes, and rare neutrophils in most animals, which was consistent with endothelialitis [65] (Figure 2g.3). Infiltrates of inflammatory cells decreased by day 14 when only few lymphocytes, plasma cells, and histiocytes surrounding the alveolar ducts were observed (Figure 2c). Alveoli adjacent to terminal bronchioles were multifocally lined by cuboidal cells that resembled bronchiolar epithelium (alveolar bronchiolization) [46,66] (Figure 2g.4). There was no evidence of fibroplasia or reparative fibrosis (Figure 2c).

Intestines and brains showed no gross nor histologically detectable lesions (Appendix A).
Figure 2SARS-CoV-2 infection results in severe but rapidly resolving pulmonary lesions in male and female Syrian hamsters. (**a**–**c**) Representative images of Syrian hamster lungs collected at 2, 6, and 14 dpi and those of mock animals. The images show H&E-stained sections. Scale bar = 200 μm. All animals were analyzed; representative images are shown. (**d**) Cumulative score of lung pathology for nine histopathological assessments in male and female hamsters (Appendix A); mean values ± SD are represented. Wilcoxon–Mann–Whitney test for males vs. females: 2 dpi *p* = 0.86; 6 dpi *p* = 0.31; 14 dpi *p* = 0.72. (**e**,**f**) Histopathological scores of intra-alveolar inflammatory cell infiltration (**e**) and perivascular/alveolar edema (**f**) in male and female hamsters (Appendix A; all animals were analyzed). Mean values ± SD are represented. Wilcoxon–Mann–Whitney test of mock animals vs. females of total histopathological score (2 dpi *p* = 0.0007; 6 dpi *p* = 0.0003; 14 dpi *p* = 0.0003), intra-alveolar inflammatory cell infiltration (2 dpi *p* = 0.0003; 6 dpi *p* = 0.0003; 14 dpi *p* = 0.0003), and perivascular/alveolar edema (2 dpi *p* > 0.99; 6 dpi *p* = 0.0003; 14 dpi = 0.52). Wilcoxon–Mann–Whitney test of mock animals vs. males of total histopathological score (2 dpi *p* = 0.0007; 6 dpi *p* = 0.0003; 14 dpi *p* = 0.0003), intra-alveolar inflammatory cell infiltration (2 dpi *p* = 0.0003; 6 dpi *p* = 0.0003; 14 dpi *p* = 0.0003), and perivascular/alveolar edema (2 dpi *p* = 0.19; 6 dpi *p* = 0.0003; 14 dpi > 0.99). Wilcoxon–Mann–Whitney test for males vs. females of total histological score (2 dpi *p* = 0.86; 6 dpi *p* = 0.31; 14 dpi *p* > 0.72), intra-alveolar inflammatory cell infiltration (2 dpi *p* > 0.99; 6 dpi *p* = 0.21; 14 dpi *p* > 0.99), and perivascular/alveolar edema (2 dpi *p* = 0.40; 6 dpi *p* = 0.12; 14 dpi > 0.99). Note: * indicates a statistically significant comparison. (**g**) 1: Severe bronchiolar epithelium and pneumocyte II hyperplasia at 6 dpi; nuclei of proliferating cells were frequently megalic with prominent nucleoli and numerous mitotic figures; 2: a syncytial epithelial cell containing multiple 2–4 µm amphophilic round cytoplasmic viral-like inclusions in a male hamster at 6 dpi; 3: lymphomonocytic endothelialitis and perivascular cuffing in a pulmonary venule at 6 dpi; 4: alveolar bronchiolization with acinar formations and few interstitial lymphoplasmacytic infiltration at 14 dpi. The images show H&E-stained sections. Scale bar = 50 μm.
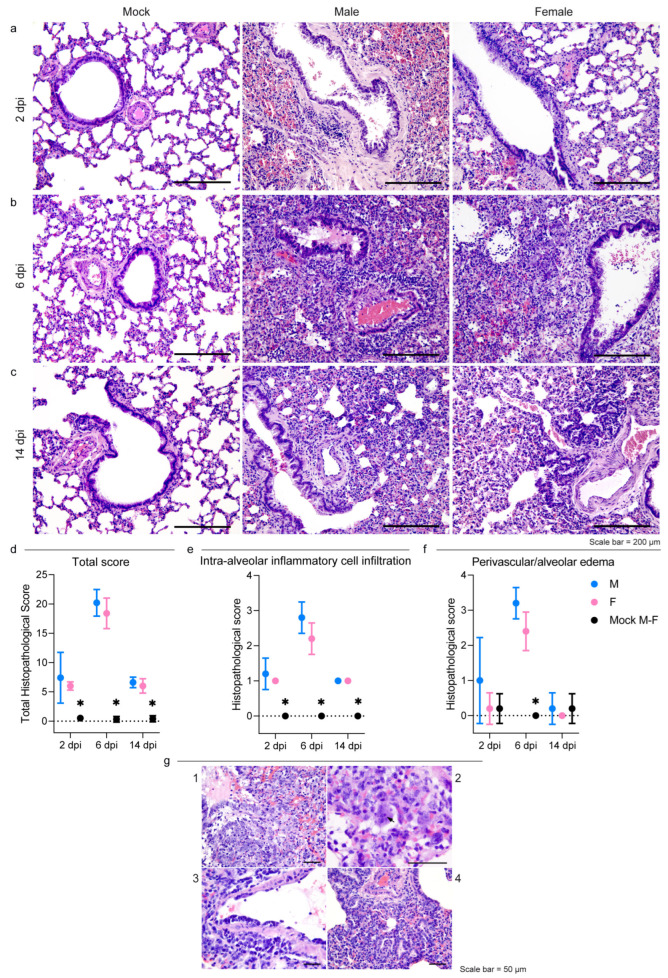



### 3.3. Host Response to SARS-CoV-2 Infection

With the purpose of studying male and female host responses to SARS-CoV-2, we performed an RNA-Seq analysis on lungs, intestines, brains, and peripheral blood mononuclear cells (PBMCs) at three different time points. The comparison of the expression profiles of all tissues from infected and mock individuals of the same sex and from the time point allowed us to quantify and describe the host’s response in terms of differentially expressed genes (DEGs) (Appendix A). In the lungs, the response began at 2 dpi, reached an apex at 6 dpi, and was still persistent at the latest time point with no substantial differences between male and female hamsters. Females PBMCs exhibited the same parabolic curve observed for the lungs, while males elicited a stronger systemic response that involved more than 2000 DEGs throughout the study (Figure 3a,b; Appendix A). In the intestines and brains, consistent with the viral presence and replication, we observed that the host’s response was far less marked. In both organs, females and males followed opposite trends: the number of DEGs increased in females and decreased in males (Figure 3b). We then employed the Gene Ontology (GO) resource to investigate the biological processes enriched in the SARS-CoV-2-infected Syrian hamsters (Appendix A). Except for the intestines of males, which showed many enriched GO terms, the number of enriched processes followed the same trend for DEGs in all tissues and both sexes (Figure 3c).

To better investigate how the Syrian hamsters responded to SARS-CoV-2, we focused on the GO terms associated with the immune response and correlated biological functions. In the lungs, some GO terms showed the same pattern of enrichment across sexes and were activated in all the infected hamsters at 2 dpi (e.g., “cellular response to type I interferon”), 6 dpi (e.g., “T cell receptor signalling pathway” and “response to interferon-gamma”), or both (e.g., “inflammatory response”, “defense response to virus”, “activation of immune response”, and “cytokine-mediated signalling pathway”) (Figure 4a). On the other hand, at 6 dpi some GO terms were exclusively enriched in males (e.g., “angiogenesis” and “negative regulation of immune system process”) or exclusively in females (“regulation of B cell differentiation”) (Figure 4a).

In both the intestines and brains, we were unable to find a clear inflammatory pattern in response to the infection with SARS-CoV-2 (Figure 4b,c).

In the intestines, we found few GO terms related to the immune system. At 2 dpi, “positive regulation of innate immune response”, “defense response to virus”, and “cellular response to interferon-alpha” were enriched in both sexes, while “cellular response to interferon-beta” and “toll-like receptor signalling pathway” were specifically enhanced in males. Only three GO terms of very general means (e.g., “inflammatory response”) were enriched at 6 dpi, while at 14 dpi we detected GO terms related to lesion recovery such as “wound healing” and “tissue regeneration”. At this time point, we noted few sex-specific enriched terms such as “positive regulation of T cell differentiation”, “lymphocyte differentiation”, and “B cell activation” in males and “antigen receptor-mediated signalling pathway” and “chemokine-mediated and cytokine-mediated signalling pathways” in females (Figure 4b).

In the brains, few GO terms were enriched in both sexes exclusively at 2 dpi; these included “activation of immune response”, “complement activation”, “defense response to virus”, and “cellular response to interferon-alpha and -beta” (Figure 4c).

As a major novelty of this study, we analyzed the immunological profile of PBMCs to investigate the Syrian hamsters’ systemic activation of their immune systems to search for potential similarities with severe human COVID-19 cases. We observed the activation of the immune response in both sexes at all three time points as expressed by the longitudinal enrichment of related GO terms such as “cytokine/chemokine-mediated signalling pathway”, “regulation of lymphocyte activation”, “inflammatory response”, “programmed cell death”, and “defense response to virus” (Figure 5a). Other GO terms enriched in both sexes during the experiment at any time point included “complement activation”, “antigen processing and presentation of peptide antigen via MHC class I”, “positive regulation of innate immune response”, and “toll-like/pattern recognition receptor signalling pathway”. Some GO terms were enriched in a sex-specific manner; these included “regulation of autophagy” and “lymphocyte differentiation in males” or “cellular response to interferon-alpha” and “alpha-beta T cell activation” in females. In particular, we observed major differences between females and males at 14 dpi when 98% of the upregulated genes (2676/2723) were male-specific and 60% of downregulated genes (401/672) were female-specific (Figure 5b; Appendix A).
Figure 3RNA-Seq global expression profiles. (**a**) Volcano and MA plots showing differential expression analysis results for lungs/intestines/brains and PBMCs, respectively (blue, upregulated; red, downregulated; grey, not significant). A DEG was significant in a comparison when Log2FC ≤ −1 or Log2FC ≥ 1and FDR < 0.05. For lungs, intestines, and brains: x axis = Log2FC; y axis = −LogFDR. For PBMCs: x axis = Log mean expression; y axis = Log2FC. (**b**) Number of DEGs for every comparison of infected vs. mock found in the differential expression analysis; up- and downregulated genes are shown. See also Appendix A for the raw numbers of DEGs. (**c**) Number of enriched GO terms for every comparison of infected vs. mock found in the Gene Ontology enrichment analysis. See also Appendix A for the GO term numbers. Infected female and male lungs and intestines: 2, 6, and 14 dpi n = 3 each; infected female and male brains: 2 dpi n = 3, 6 dpi n = 2, and 14 dpi n = 3 each; infected female and male PBMCs: a pool of 5 animals’ blood was analyzed at each time point.
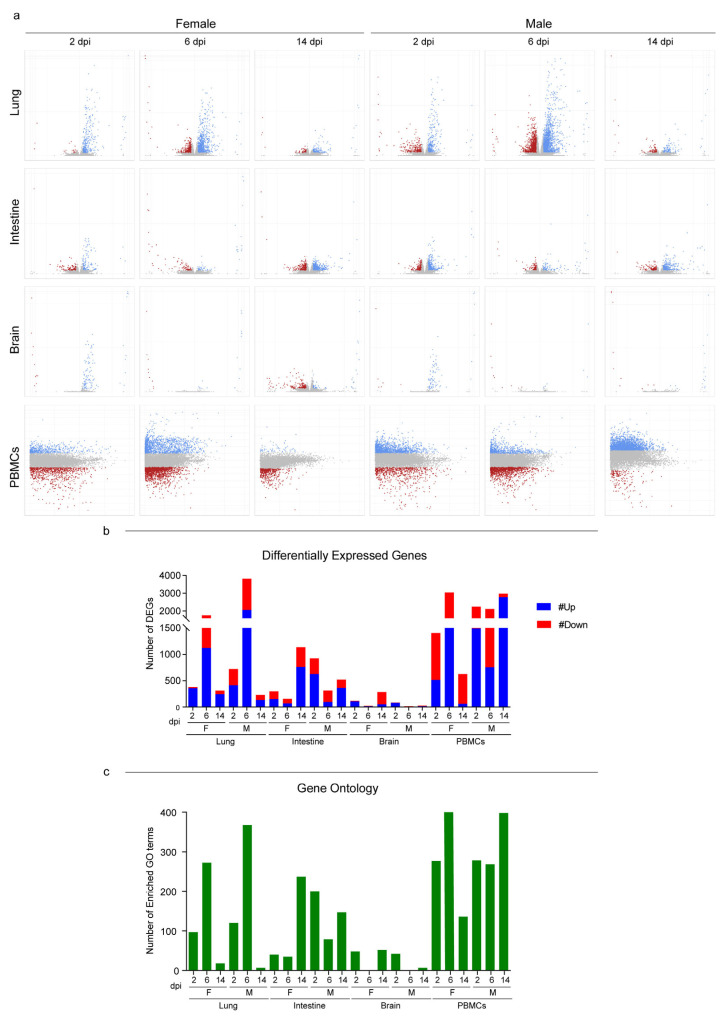

Figure 4Transcriptomic profile of SARS-CoV-2-infected male and female Syrian hamsters. (**a**) Dotplot representing the most specific enriched Gene Ontology (GO) terms related to immunity in lungs. (**b**) Dotplot representing the most specific enriched GO terms related to immunity in intestines. (**c**) Dotplot representing the most specific enriched GO terms related to immunity in brains. Statistically significant enrichments (FDR < 0.05) are presented, and the −LogFDR is shown. The absence of the dot means that the indicated GO term was not enriched in that particular sample.
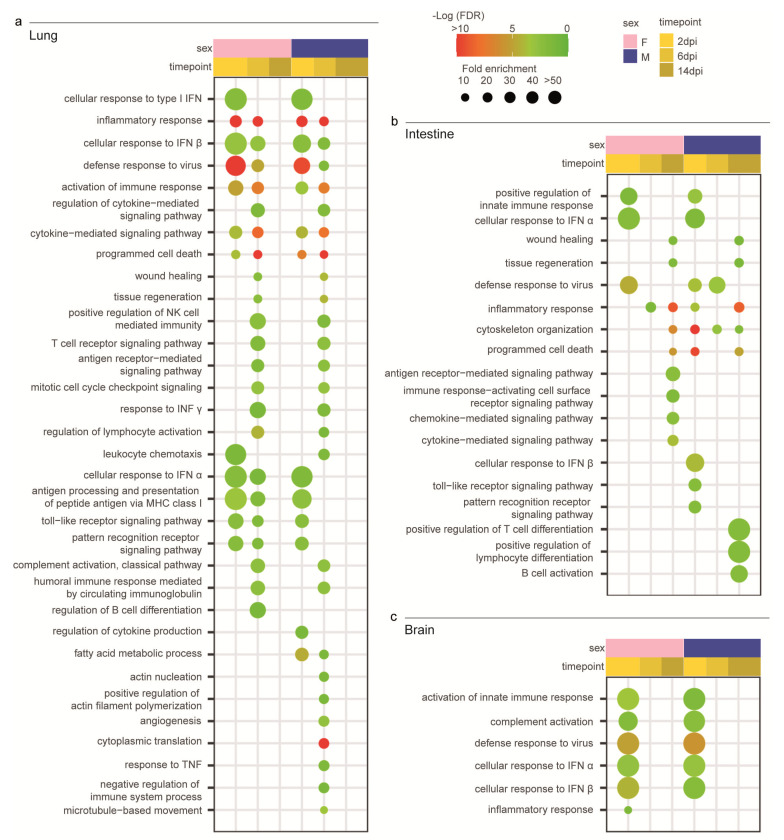

Figure 5Transcriptomic profile of SARS-CoV-2-infected male and female PBMCs. (**a**) Dotplot representing the most specific enriched GO terms related to immunity in PBMCs. Statistically significant enrichments (FDR < 0.05) are presented, and the −LogFDR is shown. (**b**) Scatterplot representing the Log2FC of male and female DEGs in PBMCs at 14 dpi. DE = differentially expressed. A DEG was significant in a comparison when Log2FC ≤ −1 or Log2FC ≥ 1and FDR < 0.05. The absence of the dot means that the indicated GO term was not enriched in that particular sample.
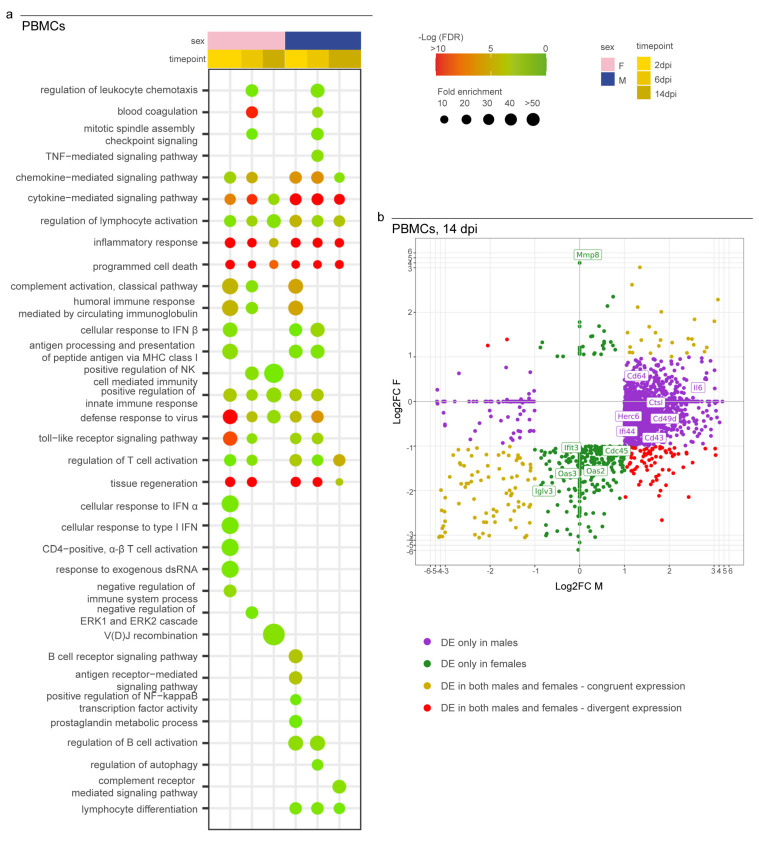



### 3.4. Syrian Hamster as Immunological Model for COVID-19

To investigate whether the hamsters displayed the typical immunological profiles described in human COVID-19 lungs, we evaluated the expression levels of 100 genes associated with a severe human condition in previous studies [13,66,67,68,69,70,71,72] (Figure 6a). The Syrian hamsters activated an interferon-I (IFN-I)-mediated cell-specific response to the virus at 2 dpi as shown by the upregulation of many interferon-stimulated genes (ISGs). This included genes that coded for IFIT proteins (e.g., *Ifit2* and *Ifit3*), members of the OAS family (e.g., *Oas1*, *Oas2*, *Oas3*, and *Oasl*), interferon regulatory factors (e.g., *Irf7* and *Irf9*), and several genes involved with cellular mechanisms of antiviral response (e.g., *Ddx60*, *Parp12*, and *Parp14*). The specific immune response increased in both sexes at 6 dpi and showed the upregulation of 58 and 50 out of 100 target genes for males and females, respectively. SARS-CoV-2-infected animals promoted immune cell recruitment with complement activation, immunoglobulin-mediated response, and strong upregulation of pro-inflammatory cytokines (e.g., *Ccl2*, *Ccl3*, *Cxcl10*, *Il6*, and *Ifnγ*); moreover, we observed the activation of the genes involved in monocyte (*Cd33*, *Cd16*, and *Siglec1*) and T-cell activation (*Tbx21*, *Cd40lg*, *Cd4*, *Cd8a*, and *Cd8b*). At 6 dpi, males also upregulated genes associated with active neutrophil recruitment (e.g., *Mmp9*, *Cd11c*, *Fut4*, and *Elane*) and angiogenesis (e.g., *Mmp3*, *Thbs1*, and *Angptl4*) (Figure 6a). Interestingly, male hamsters downregulated both the SARS-CoV-2 receptor *Ace2* and its receptor *Agtr1* genes. Of note, immunofluorescence staining for ACE2 expression confirmed the sex-specific reduction in the receptor in lung tissues compared to mock controls (Figure 6b). Hamsters of both sexes almost completely shut down the specific pulmonary immune response by day 14, and no individuals perpetuated the immune exasperation and inflammation typical of severe COVID-19.

Among the 32 key immunological genes associated with a severe COVID-19 systemic pathology in humans, 24 were differentially expressed in the hamsters’ PBMCs in at least one case (Figure 6c). Our results presented a male-biased upregulation of genes associated with immature neutrophil activation (e.g., *Cd49d*, *Cd274*, *Tlr4*, and *Cd43*) and pro-inflammatory cytokines associated with the cytokine storm (*Il1β*, *Il6*, and *Tnf*). In this context, the longitudinal monitoring of pro-inflammatory IL-1β and IL-6 revealed their low release in the serum in response to SARS-CoV-2 infection, and there was an exclusive increase in the circulating levels of IL-1β in male hamsters at 14 dpi (Figure 6d and Appendix A).
Figure 6Transcriptomic analysis highlighted differences in males’ and females’ systemic responses to SARS-CoV-2 infection. (**a**) Heatmap (Log2FC values of the performed comparisons) of selected genes related to the immune system in the lungs. A DEG was significant in a comparison when Log2FC ≤ −1 or Log2FC ≥ 1and FDR < 0.05. (**b**) Immunofluorescence staining for the SARS-CoV-2 receptor ACE2 in infected and control male and female lungs at 6 dpi. All animals were analyzed; representative images are shown. Scale bar = 25 µm. (**c**) Heatmap (Log2FC values of the performed comparisons) of selected genes related to the immune system in the PBMC transcriptome. Note: * indicates the gene name for hamsters in case it differed from the human ortholog. (**d**) Singleplex ELISA levels of IL-1β (pg/mL) for mock and infected male and female hamsters. Wilcoxon–Mann–Whitney tests of mock or SARS-CoV-2-infected males vs. females; mean ± SEM is represented (* indicates a statistically significant comparison).
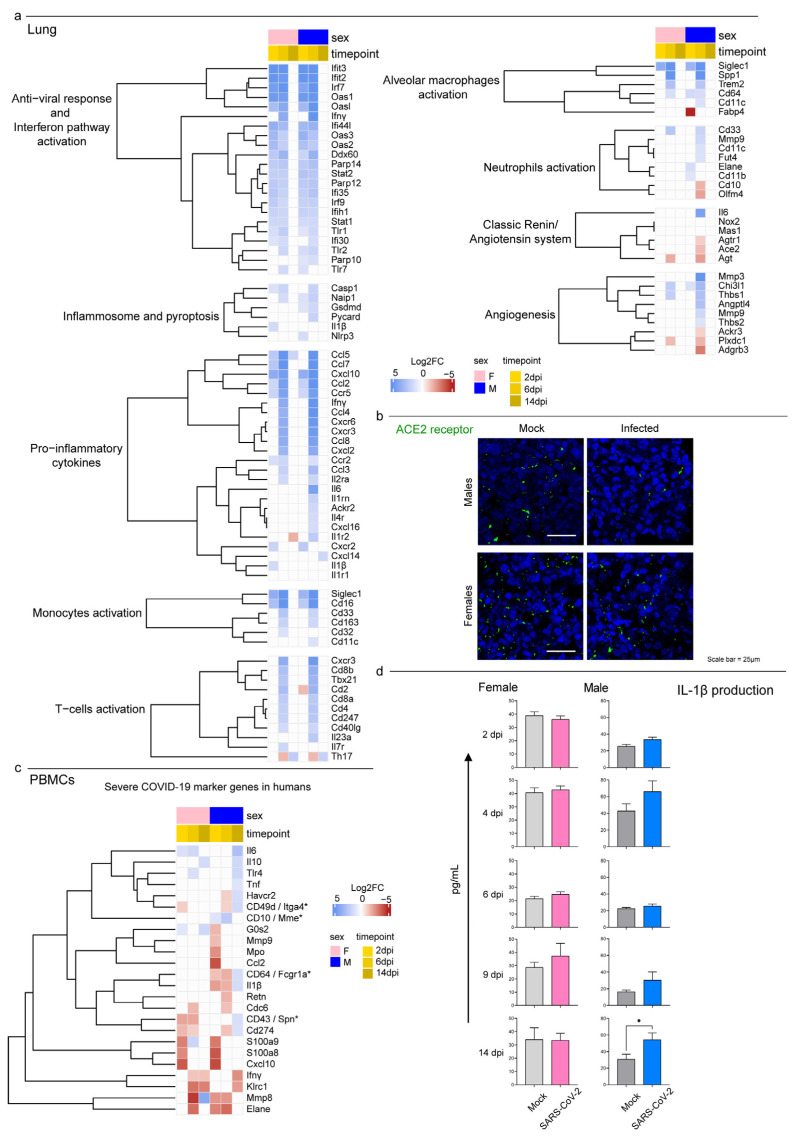



## 4. Discussion

Following the declaration of the COVID-19 pandemic by the WHO in March 2020, both the scientific community and health authorities were on the front line of the development of control measures to limit the spread of the infection and mitigate disease severity. To achieve this goal, translational animal models were used to elucidate the pathogenesis of the disease and to rapidly assess the efficacy of prophylactic and therapeutic agents. However, in order for scientists to select the best animal models for their studies, it is crucial to characterize in which way a species can mimic the host–pathogen relationship between humans and SARS-CoV-2. In this study, we provided a comprehensive description for Syrian hamsters, which were extensively used as an animal model long before the emergence of SARS-CoV-2 to study other emerging zoonotic disease vectors such as bunyaviruses, arenaviruses, henipaviruses, flaviviruses, alphaviruses, filoviruses, and the coronaviruses SARS-CoV and MERS-CoV [73]. We performed experimental infections using the SARS-CoV2 B.1.1.7 strain or the Alpha VOC isolated in Italy. In our study, we successfully infected all of the hamsters and detected the RNA of SARS-CoV-2 in tissues and oropharyngeal swabs from day 2 to 14 and a specific antibody response by day 6, which supporting earlier evidence [38,40,42]. In addition, we confirmed the presence of the antigen within the pulmonary tissue up to 14 dpi via immunofluorescence by using a specific antibody directed toward the spike protein. Unfortunately, our choice to store samples only in RNA later and formalin, which was driven by the need to secure the quality of the molecular and histological analyses, hampered any isolation attempt to test the viability of the virus within different tissues. However, based on other studies, no viable virus was present in the lungs after day 7 [42], which led us to assume that our findings were associated with the presence of residual RNA/proteins rather than with an active infection. Indeed, we implemented a specific immunofluorescence technique to detect dsRNA, which is a replicative intermediate of many RNA viruses (including coronaviruses) [49,51]. Interestingly, we found positive staining for dsRNA only in male lungs at 6 dpi, which suggested a sex-related difference in the replication rate. However, such a difference did not translate into an evident increase in the total virus amount using ddPCR. In this sense, paired immunofluorescence and molecular investigations performed at intermediate timepoints might have helped to elucidate the dynamics of viral infection and replication within the pulmonary tissues of female and male hamsters.

The SARS-CoV-2-infected hamsters developed moderate to severe bronchointerstitial pneumonia, which mimicked histological patterns observed in COVID-19 patients (i.e., diffuse alveolar damage (DAD), interstitial and intra-alveolar influx of macrophages/neutrophils, and pulmonary vascular endothelialitis) as previously described [38,40,43,44,74]. Compared to the pneumonia described in human COVID-19, the DAD was milder and unevenly distributed; in addition, we did not notice the formation of hyaline membranes. While some studies confirmed DAD in hamsters using the original USA-WA1/2020 isolate [39], our findings were in line with current literature that supports the evidence of a milder disease in this animal model, especially when considering that most human findings were derived from patients who died of severe COVID-19 [38,42,75,76,77].

Despite lung damage, the hamsters showed no clinical signs but did show a significant loss in body weight that resolved spontaneously by day 14 after the infection. This result was consistent with previous reports [31], although a few studies also described symptoms such as lethargy, ruffled fur, a hunched back posture, and rapid breathing [38,39], a difference that might have been related to the virus (i.e., the titer and route of inoculum or viral strain) [40,78], the hamsters (i.e., their age) [40,79], or a combination of both [40]. While it is known that prey species such as hamsters mask their sickness when they perceive a threat such as the presence of humans [80], these data suggested that the disease in hamsters mostly resembles that found in humans with mild COVID-19 symptoms. In humans, severe COVID-19 is associated with tissue damage due to an exacerbated inflammatory response [81] and multiorgan failure as a secondary effect of systemic activation and exhaustion of the immune system [67,82] or is due to viral spread outside the respiratory system [83,84]. In our study, we investigated the viral spread and the hamsters’ immune responses at the local and systemic levels in order to evaluate the differences between our animal model and severe cases of COVID-19 in humans.

The hamsters mostly responded to SARS-CoV-2 in the lungs within the first week followed by subsequent silencing by the end of the experiment that combined the recovery from the clinical disease, the clearance of the infection, and the repair of pathological lesions. Most DEGs and GOs were associated with the immune response and related biological functions (including the activation of IFN-I alpha and beta) as previously described [42,44]. These molecules were crucial to an effective antiviral response because they counteracted viral replication in infected cells and cell-to-cell spread, enhance antigen presentation, and promoted the development of the adaptive immune response [85,86]. Despite the induction of interferon dampening after an infection with SARS-CoV-2 compared to other viruses such as influenza A [14,87], IFN-I signaling influences the severity of COVID-19 in humans. Alterations in TLR3-dependent and TLR7-dependent type I interferon induction, the presence of autoantibodies to interferon, and the general reduced induction of local and systemic interferon responses against SARS-CoV-2 infection lead to severe manifestations [14]. Indeed, a restricted IFN-I response might promote longstanding active viral replication, excessive production of pro-inflammatory cytokines, and influx of neutrophils and monocytes, which act as further sources for pro-inflammatory mediators and promote greater tissue damage [81]. In this context, it is likely that the early and powerful induction of IFN-I-related genes that we described in the hamsters promoted fast viral clearance in the lungs and tissue structure restoration, thereby preventing severe manifestations of the disease and the systemic spread of the virus in this animal model. Our data showed that in a manner similar to humans, the hamsters also responded to the infection with local inflammation, recruitment of immune cells, activation of the complement and immunoglobulin-mediated response, and release of pro-inflammatory cytokines. However, such a response was contained in this animal model and shut down by day 14 after the infection. Furthermore, the PBMC RNA-Seq data showed a modest systemic response in the hamsters that resolved within two weeks with the activation of the interferon pathway, innate cell recruitment, and the activation of lymphocytes B and T and an immunoglobulin-mediated immune response. A modest increase in the circulating levels of pro-inflammatory cytokines further corroborated previous studies [88,89] and highlighted another crucial difference in patients suffering from complicated COVID-19 that present almost threefold higher levels of the pro-inflammatory cytokine IL-6 compared to patients with an uncomplicated form of the disease [90]. Overall, our data suggested that the hamsters did not suffer from any dysregulation of the immune system that might determine severe COVID-19 in humans.

Consistently with the low systemic activation of the immune system that promotes tissue damage in peripheral districts in humans, we discovered that there were no histopathological lesions in the intestines and brains of the hamsters. In addition, our ddRT-PCR data supported other studies in showing the limited spread of SARS-CoV-2 outside the respiratory tract in this species [40]. These data were in line with the current literature and included minimal spread to the kidneys and heart [42]. The lower or absent systemic infection in the hamsters compared to humans, in which the virus can spread to the digestive tract, the brain, the heart, the kidneys, the sweat glands of the skin, and the testicles [83,84], further explained the fewer complications seen in this model. Interestingly, we found positive immunofluorescence staining for the spike and dsRNA, thus supporting replication of the virus in the intestines, while the transcriptomic analyses showed a weak and generic immune response. While the lack of studies on the transcriptome of human intestines during COVID-19 prevented us from making significant comparisons with our animal model, the infection of human small intestinal organoids resulted in much higher transcriptomic signals [91]. On the other hand, the minimal alterations shown in our analyses could simply have resulted from enterocyte sloughing that followed fasting and weight loss.

In our study, all data supported the assumption that infection with SARS-CoV-2 has more severe consequences in male hamsters. Indeed, males developed more diffuse and severe lung lesions that were characterized by higher scores of infiltration of inflammatory cells and edema, which may have resulted in the more obvious pathological manifestations. Thanks to the combination of several approaches, our study allowed us to investigate the possible causes and consequences of such a difference. Of note, we found that in the lungs, males displayed a higher differential expression of genes associated with both activated neutrophils and alveolar macrophages as well as with the release of pro-inflammatory cytokines such as *Il-6*, *Cxcl10*, and *Ifnɣ* that are associated with ARDS. This sex-based difference has been evidenced in human COVID-19 cases [92,93,94,95], but it had not been previously reported for animal models where the transcriptome of infected hamsters was mainly investigated using RT-qPCR rather than RNA-Seq analysis [79,96,97]. Another peculiarity of male hamsters that stood out from our data was the differential expression of genes that promote angiogenesis (e.g., *Mmp3*, *Thbs1*, and *Angptl4*), which might explain the sex-driven differences in the pulmonary lesions. Finally, the male hamsters downregulated both *Ace2* and its receptor *Agtr1* on day 6, a feature that we were able to identify using transcriptomic analyses and to confirm through immunofluorescence, which showed a decreased level of the receptor within the pulmonary tissue between non-infected and infected animals. As the receptor is endocytosed together with the virus during cellular infection, this difference might be due to a higher level of infection and replication of SARS-CoV-2 or to an increased cell death and apoptosis in males. In this context, we found a general enrichment of the programmed cell death process in lungs between 2 and 6 dpi, but there were no differences between females and males (Figure 4a). On the other hand, we observed a high viral load via ddRT-PCR and a peculiar staining for dsRNA in the lungs of male hamsters, which corroborated a possible link between the regulation of *Ace2* and the replication of the virus. Other than being a SARS-CoV-2 cellular receptor, ACE2 has the physiological function of inactivating angiotensin II (AII) molecules produced by ACE, which is known for its vasoconstrictive activities and—crucially—for acting as a potent pro-inflammatory cytokine [81]. As a further sign, an increased level of AII can also exacerbate IL-6 signaling. In this context, several cytokine storm cytokines [82,98] such as *Il-6, Il-1ꞵ*, and *Tnf* as well as genes associated with immature neutrophil activation (e.g., *Itga4*, *Cd274*, and *Spn*) were specifically upregulated in PBMCs from male hamsters only. Similarly, males showed a peculiar increase in serum levels of IL-1β at 14 dpi that was not observed in females, thus suggesting a possible re-exacerbation of the systemic inflammation.

These results further corroborated a sex-mediated difference in the pathology of COVID-19 in hamsters that could provide useful insights into understanding similar evidence in humans. Indeed, studies worldwide have supported that more men than women require intensive care or succumb to the disease [16]. While it has been suggested that social and behavioral differences between genders might influence the progression of COVID-19, our data supported the role of the biological sex. Finally, the longitudinal assessment of oropharyngeal swabs demonstrated that while they showed similar kinetics, males eliminated more virus, which also suggested sex-driven differences in the epidemiology of the pandemic.

## 5. Conclusions

Our study provided a comprehensive evaluation of the Syrian hamster as animal model for COVID-19. Overall, we confirmed that the infection with SARS-CoV-2 showed similar pathways both in humans and hamsters, the latter of which proved to be an excellent model to test the efficacy of prophylactic biologicals such as vaccines and to quickly assess the phenotypic changes of new VOCs [31]. As a matter of fact, our clinical and histopathological data suggested a decreased pathogenicity of the Alpha VOC compared to the original strain, which was found to induce DAD and cause more severe symptoms in the hamster [38,39]. More recently, the experimental infection of hamsters was also effective in showing the decreased pathogenicity of the Omicron VOC upon emergence, which confirmed preliminary epidemiological data obtained from the human population [43]. In addition, Syrian hamsters have been successfully used to investigate the transmission of SARS-CoV-2 between cohoused contact hamsters [99], which is a feature that often requires the use of ferrets, a model that is associated with higher costs and logistical challenges [31]. While the fast recovery of the animals and clearance of the virus observed in our study would suggest otherwise, the Syrian hamster was also found to be an effective model to evaluate the long-term effects of SARS-CoV-2 infection that are collectively known as “long-COVID syndrome” or post-acute sequalae of COVID-19 (PASC) [100,101], which is still a very heterogeneous and poorly understood condition in humans. Indeed, Frere et al. [42] showed that hamsters infected with the original WA1/2020 strain displayed transcriptional alterations in the striatum brain and persistent inflammatory profile in the olfactory bulb more than one month after infection, which could be associated with cognitive impairments, depressive disorders, and behavioral changes seen in patients suffering from long COVID [42].

On the other hand, our study underlined that hamsters only mimic mild to moderate cases of COVID-19 and neither replicate the exacerbation of the immune response nor the systemic spread. So far, the hamster model has failed to recapitulate the most severe human symptoms regardless of the variant used in the experiments, which suggests that our data could be at least partially extrapolated outside the specific case of the Alpha VOC [31]. In this context, hamsters should be used with caution in more detailed studies on COVID-19 and to evaluate therapeutic agents that dampen the immune response. Among other species, African green monkeys are currently the best model to mirror severe disease manifestations seen in humans (notably ARDS) [31]. Most recently, mouse models engineered or engrafted with human tissue also proved to be effective in mimicking the physiological features of human infection, thereby allowing for detailed pathological studies [31]. However, most of these models have several constraints for widespread use such as a lower availability, higher costs, and difficulties in handling and management within experimental settings. It is also crucial to mention the increasing role of alternative models, among which organoids obtained by using human stem cells have emerged as powerful tools that are able to bridge the gap between cell lines and animal models and to scale up to high-throughput screens [102]. These systems have been widely used to study SARS-CoV-2 infection (including viral tropism, the host response, and drug discovery for different VOCs) [102]. Future perspectives include the production of organoids with diverse genetic backgrounds to explore the impact of host genetics on disease progression and the responses to vaccines and therapeutic agents [103].

As a final note, we were able to observe a significant difference between female and male hamsters that should be taken into account when designing any experimental study. While this feature is possibly not peculiar to SARS-CoV-2 infection, the sex biases of animal experiments have long represented a critical aspect of translational medicine [103]. Fortunately, researchers, funders, and policy makers unanimously acknowledge the need for a change; research projects that include both sexes and analyses of data according to gender—as in the present study—are becoming increasingly popular. In turn, we believe that animal models will progressively become important not only to describe the pathological pathways of a disease but also to grasp differences related to biological sex.

## Data Availability

The RNA-Seq raw data generated in the present study were deposited in SRA under accession number PRJNA839918.

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
