# Peer review of "Host Response of Syrian Hamster to SARS-CoV-2 Infection including Differences with Humans and between Sexes"

_viruses, 2023, doi:10.3390/v15020428_

Round 1
Reviewer 1 Report
This study characterized SARS-CoV-2 infection in Syrian hamster by measuring weight change, clinical signs, viral replication, receptor profiling and host immune response in four different organs (lungs, intestine, brain and PBMCs). Data of this study showed both male and female hamsters are susceptible to the infection and develop a disease similar to human COVID-19. Pathological results showed moderate to severe pulmonary lesions, inflammation and recruitment of the immune system in lungs. Males sustained higher viral shedding and replication in lungs, suffered from more severe symptoms and histopathological lesions.
1. No plaque assay was performed to detect viable virus in organs harvested from infected animals. The low copy numbers of virus RNA detected in intestine and brain may not be viable virus, which explains why not much tissue damage was observed in these two organs.
2. Another possibility of observed low copy numbers of viral RNA in intestine and brain might be cross-contamination during organ collection. Authors need to provide detailed procedure of necropsy for this study.
3. Virus strain information need to be provided, such as what cell line for virus amplification, what media, which passage of virus, etc.
4. In Fig. 1d, sample sizes for two gender groups are too small to conduct statistical comparison. Especially for brain samples at 6 dpi.
5. How the challenge dose was determined? Why it is 8X10^4?
6. Fig. 1e needs to have a panel for phase contrast to show structure of lungs. In addition, the same staining should be done with intestine and brain.
7. In Fig. 1a, it would be better to present percent of body weight change using original individual body weight as 100%.
8. In page 6 line 253, authors state “… in one case, contained 2-4 μm amphophilic round cytoplasmic inclusions consistent with viral like particles…”. This statement needs to rephrase. Viral size of SARS-CoV-2 is only 70 – 90 nM, much smaller than 2-4 microns.
9. Font in figures need to be enlarged for better visualization. For example, Fig. 3b and 3c X-axis label, Fig. 4a “sex,” “timepoint” label.
1. The introduction of viral strain could be moved from Discussion into Introduction.
Reviewer 2 Report
This manuscript is well written and easy to understand. Although the data presented here is not entirely novel and the hamster model is widely used/studied by scientific community across academia/industry, the conclusions in this manuscript are generally supported by experiments.
This reviewer has the following concerns-
1. lines 450-452, the authors say hamsters in their developed pneumonia mimicking histo patterns of covid patients (including diffuse alveolar damage or DAD) however they didn't observe hyaline membrane formation which is characteristic of DAD in humans. Alveolar damage was milder unlike those observed in humans. They need to address these discrepancies.
Hyaline membrane formation has been reported in hamster models (Choudhary et. 2022, Francis 2021). Authors need to cite these references and discus them briefly.
Francis ME, Goncin U, Kroeker A, et al. SARS-CoV-2 infection in the Syrian hamster model causes inflammation as well as type I interferon dysregulation in both respiratory and non-respiratory tissues including the heart and kidney. PLoS Pathog. 2021 Jul 15;17(7):e1009705.
Choudhary S, Kanevsky I, Yildiz S, et al. Modeling SARS-CoV-2: Comparative Pathology in Rhesus Macaque and Golden Syrian Hamster Models. Toxicol Pathol. 2022;50(3):280-293. doi:10.1177/01926233211072767
2. In this study, hamsters recovered within 14 days which is in line with many other previous reports. However, a recent study reported lasting and systemic perturbations after recovery (Frere et al. 2022; see fig 3). Authors need to discuss hamster models in light of recent publications to show its usability in studying long COVID.
Frere JJ, Serafini RA, Pryce KD, et al. SARS-CoV-2 infection in hamsters and humans results in lasting and unique systemic perturbations after recovery. Sci Transl Med. 2022;14(664):eabq3059. doi:10.1126/scitranslmed.abq3059
3. Outside respiratory system, the focus was limited to intestinal and nervous systems. Any reason why authors limit themselves to these two systems? Did authors evaluate the heart and kidneys? Studies have shown these organs as targets too. (Francis et al; ferre et al)
Francis ME, Goncin U, Kroeker A, et al. SARS-CoV-2 infection in the Syrian hamster model causes inflammation as well as type I interferon dysregulation in both respiratory and non-respiratory tissues including the heart and kidney. PLoS Pathog. 2021 Jul 15;17(7):e1009705.
Frere JJ, Serafini RA, Pryce KD, et al. SARS-CoV-2 infection in hamsters and humans results in lasting and unique systemic perturbations after recovery. Sci Transl Med. 2022;14(664):eabq3059. doi:10.1126/scitranslmed.abq3059
4. Apart from Syrian hamsters there are other hamster disease models available (e.g., Tg hACE2, Chinese, Roborski dwarf). Authors need to discuss them briefly in the introduction section to show why Syrian hamsters are best model to use.
Reviewer 3 Report
In this work by Castellan et al., the authors performed a deep characterization of the pathophysiology and immune signatures of SARS-CoV-2 infection in the golden Syrian Hamster animal model, specially focusing on sex-related differences and their similarities with COVID-19 in humans.
The transcriptomics part is well performed and some of the results are interesting and in line with previous findings. As the authors clearly state in the introduction, a deep characterisation of golden Syrian Hamsters as a model of SARS-CoV-2 is relevant, since it is one of the main models used to screen for antivirals and test novel vaccines, not only for SARS-CoV-2, but also for other emerging viruses.
In the introduction the authors cite relevant studies related to the field, but especially focus on early publications from 2020 and 2021. Considering how fast our understanding of the SARS-CoV-2 pathophysiology and viral fitness is evolving, and the high number of manuscripts being published, the introduction feels outdated.
In line with this previous comment, the authors used the variant B.1.1.7 or Alpha to perform the experimental infection in golden Syrian Hamsters. Even though the authors explain a bit in the discussion the relevance of this variant, since it was the first “variant of concern” reported by the WHO, the alpha variant has been displaced by other variants for more than a year and a half. That is why I think that this work has lost its momentum. For instance, I think this work would have more impact if the ancestral wildtype virus or one of the omicron sublineages had been employed. I understand this work may have been performed in 2021, but I think the manuscript would clearly benefit from a wider discussion on how their findings can be extrapolated to the other SARS-CoV-2 variants, or how this model may or may not be relevant for the study of long-covid, which is one of the current concerns of the field.
Another major concern is that the title is too general and does not reflect the specific findings of this work. While it is true that they performed a solid characterisation of some aspects of the model, I think that some of their techniques (clinical signs, shedding, pathophysiology and gene copies) are too general to be considered “in-depth”. I think that the title should include a reference to the innate immune signatures detected by the authors and the sex-related differences observed.
The authors mention in the discussion that golden Syrian Hamsters are an excellent model to study SARS-CoV-2 infection but are cautious considering their use to evaluate immune-focused therapeutic agents, which I concur and think it is a good point. But I would like a more in-depth comparison with other models for the immune signatures they identified. This is important, especially considering that there is not a “best model”, but a model better fit for an experiment or a specific analysis.
Considering the methodology, I think that the analysis performed in lungs, brain, intestines and especially PBMCs is relevant, but I also miss other relevant tissues such as Nasal Turbinate, especially considering that the experimental infection is performed intranasally. Why the authors did not include this tissue?
I understand that all groups were sex-balanced, but since this work mainly focuses on sex-related differences I think it is important to indicate the % of males/females, which I could not find in the manuscript.
The animals were monitored daily for clinical signs, but the weight was not recorded daily. My understanding for this model is that the main clinical sign to analyse disease progression is the weight, and hence the standard procedure is to weight them daily.
Considering the results, I think the quality of the images in Figures 1-3 must be improved, especially for the immunofluorescence and H/E-stained sections.
For non-pathologists, the pathology section (3.2) is written too technically. For example, what are the authors deducing with the “dark-red consolidation”? Does that refer to the lesions they observed macroscopically?
The finding that only lungs at 6 dpi of infected males were positively stained by dsRNA is surprising and the authors conclude that the virus replicates better in males according to the panel displayed in Fig. 2e. These results are not supported by qPCR, since there are no differences between males and females by qPCR. Do you have more images and a way to quantify this? A useful approach here would be the isolation of replicative virus at 6 dpi in the lungs of males and females, to confirm this hypothesis. In line with this, the authors generated a model for viral shedding based on the qPCR results, but these results do not show viral shedding but RNA shedding. For viral shedding virus isolation should be performed.
I am not sure that a Mann-Whitney test can be used for the analysis performed in Figure 2, since the histopathological data is ordinal. I think the authors should use a Pearson Chi-test.
A clarification or some references of how the authors identified the 100 genes associated to severe human condition would be useful.
The authors state that males downregulated ACE2 receptor and provided nice immunofluorescence analyses supporting that. In the discussion they mention these results could be related with a higher endocytosis of the complex virus/receptor. Since there is higher replication in males, could it also be related to cell death and apoptosis?
Minor comments
· The manuscript is missing the keywords to ease the search process in specialised databases. Please include them.
· Line 52: I would rephrase “likely emerged from animals after zoonotic cross-species transmission”. It is a fact that the virus emerged from bats after a zoonotic cross-species transmission, even though the exact transmission chain and the possible intermediate species remain unknown (10.1126/science.abp8715).
· Line 77: I would mention “immune evasion” instead of “immune protection”.
· Mismatch on the order of the reference calls (jumps from 37 at the introduction to 74 at the beginning of Mat&Met). I understand that the Mat&Met section was originally placed at the end of the manuscrit and rearrenged at some point.
· The material and methods are detailed (in Supplementary Materials), but sometimes it is difficult to find some information (reagent references, histopathology scoring, etc.), which can be found in supplementary tables. Personally, I would include this information in the Material & Method section, or at least a call that directs the readers to the right section, instead of having to navigate through all the documents.
· Line 106: I would mention both variant nomenclautres for the VOC used (B.1.1.7 and alpha).
· Line 120: I would change the sentence using “…were scored as described elsewhereref”.
· Figure 2: there is a green line on the right side of Figure 2 that should not be there.
· Figure 2 legend: a-e in bold.
· In Figure 4, please, clarify if the absence of dotplot means there is no significant difference compared to mock controls.
Round 2
Reviewer 1 Report
3. Virus strain information need to be provided, such as what cell line for virus amplification, what media, which passage of virus, etc.
Thank you for the comment. All requested information are easily trackable in the reference “Giobbe, G. G. et al. SARS-CoV-2 infection and replication in human gastric organoids. Nat. Commun. 12, 1–14 (2021)”. In detail, we used a strain of B.1.1.7 lineage (Alpha Variant of Concern) isolated in 2020 from a nasopharyngeal swab collected from a mild/severe COVID case, using VERO E6 cells (ATCC® CRL 1586™).
We slightly modified the material and methods session for easier reference.
If you used original solution from the vials that came from ATCC without dilution, virus was in supernatant of cell culture medium. The two negative controls were supposed to be inoculated using 100μl of the same cell culture medium, instead of “sterile PBS solution”.
5. How the challenge dose was determined? Why it is 8X10^4?
The challenge dose referred to the titer of our undiluted original isolate. We considered this dose as appropriate for Syrian hamster infection as included in the range of doses of SARS-CoV-2 known to give a productive infection in this animal model (Imai et al., PNAS 2020; Chan et al., Clin. Infect. Dis. 2020; Armando et al., Nat. Commun. 2022; Hansen et al., Cell Rep. 2022). This information has now been added to the manuscript.
Imai, M.; Iwatsuki-Horimoto, K.; Hatta, M.; Loeber, S.; Halfmann, P.J.; Nakajima, N.; Watanabe, T.; Ujie, M.; Takahashi, K.; Ito, M.; et al. Syrian Hamsters as a Small Animal Model for SARS-CoV-2 Infection and Countermeasure Development. Proc. Natl. Acad. Sci. U. S. A. 2020, 117, 16587–16595, doi:10.1073/pnas.2009799117.
Chan, J.F.W.; Zhang, A.J.; Yuan, S.; Poon, V.K.M.; Chan, C.C.S.; Lee, A.C.Y.; Chan, W.M.; Fan, Z.; Tsoi, H.W.; Wen, L.; et al. Simulation of the Clinical and Pathological Manifestations of Coronavirus Disease 2019 (COVID-19) in a Golden Syrian Hamster Model: Implications for Disease Pathogenesis and Transmissibility. Clin. Infect. Dis. 2020, 71, 2428–2446, doi:10.1093/cid/ciaa325.
Armando, F.; Beythien, G.; Kaiser, F.K.; Allnoch, L.; Heydemann, L.; Rosiak, M.; Becker, S.; Gonzalez-Hernandez, M.; Lamers, M.M.; Haagmans, B.L.; et al. SARS-CoV-2 Omicron Variant Causes Mild Pathology in the Upper and Lower Respiratory Tract of Hamsters. Nat. Commun. 2022, 13, doi:10.1038/s41467-022-31200-y.
Hansen, F.; Meade-White, K.; Clancy, C.; Rosenke, R.; Okumura, A.; Hawman, D.W.; Feldmann, F.; Kaza, B.; Jarvis, M.A.; Rosenke, K.; et al. SARS-CoV-2 Reinfection Prevents Acute Respiratory Disease in Syrian Hamsters but Not Replication in the Upper Respiratory Tract. Cell Rep. 2022, 38, 110515, doi:10.1016/j.celrep.2022.110515.
If you used original solution from the vials that came from ATCC without dilution, virus was in supernatant of cell culture medium. The two negative controls were supposed to be inoculated using 100μl of the same cell culture medium, instead of “sterile PBS solution”. Sterile PBS and cell culture medium are not equal, so animals inoculated with PBS cannot serve as good controls for this study.
7. In Fig. 1a, it would be better to present percent of body weight change using original individual body weight as 100%.
Thank you for the good suggestion. While it is easy to satisfy this request, it is crucial to consider that Fig 1a is not a mere representation of raw data but is actually displaying results from the spline mixed model. In the study, we used this statistical approach as it allows taking into account the correlation among repeated observations of the same hamster over time, providing statistical solid data in order to compare groups. For these reasons, the authors would prefer to maintain the original figure.
Before infection, each animal has different body weight compared to others. Using absolute body weight as a variable to compare different groups after infection cannot control this baseline difference. The benefit of using each hamster’s D0 body weight as a reference to calculate percentage of body weight change daily relative to this baseline value can normalize this baseline confounding factor and show real body weight change ascribed to the infection.
Round 3
Reviewer 1 Report
Authors stated in their response to question 3 that " the virus was isolated and grown in our laboratory...". It is a common practice to provide the information of cell line for virus amplification, media components of cell culture, passage of virus used for infection, and procedure of preparing virus for inoculation media in Methods and Materials.
